

2 # A simple global Budyko model to partition evaporation into interception and transpiration
4 **Ameneh Mianabadi[1,2], Miriam Coenders – Gerrits[2]\*, Pooya Shirazi[1], Bijan Ghahraman[1], Amin Alizadeh[1]**
**1- Ferdowsi University of Mashhad, Mashhad, Iran**
**2- Delft University of Technology, Delft, The Netherlands**
*\*Corresponding author*
**Abstract**
Evaporation is a very important flux in the hydrological cycle and links the water and energy
balance of a catchment. The Budyko framework is often used to provide a first order estimate of
evaporation, since it is a simple model where only rainfall and potential evaporation is required as
input. Many researchers have tried to improve the Budyko framework by including more physics
and catchment characteristics into the original equation. However, this often resulted in additional
parameters, which are unknown or difficult to determine. In this paper we present an improvement
of the previously presented Gerrits' model ("Analytical derivation of the Budyko curve based on
rainfall characteristics and a simple evaporation model" in Gerrits et al, 2009 WRR), whereby total
evaporation is calculated on the basis of simple interception and transpiration thresholds in
combination with measurable parameters like rainfall dynamics and storage availability from
remotely sensed data sources. While Gerrits' model was investigated for 10 catchments with
different climate conditions and also some parameters were assumed to be constant, in this study
we applied the model on the global scale and it was fed with remotely sensed input data. The output
of the model is compared to two complex land-surface models STEAM and GLEAM, as well as
the database of Landflux-EVAL. Our results showed that total evaporation estimated by Gerrits'
model is in good agreement with Landflux-EVAL, STEAM and GLEAM. Results also show that
Gerrits' model underestimated interception in comparison to STEAM and overestimated in
comparison to GLEAM, while for transpiration the opposite was found. Errors in interception can
partly be explained by differences in the interception definition that successively introduce errors
in the calculation of transpiration. Comparing to the Budyko framework, the model showed a good
performance for total evaporation estimation and the results are closer to Ol'dekop than Schreiber,
Pike and Budyko curves.
**Keywords:** Budyko curves, interception, transpiration, remote sensing, evaporation



# 1 Introduction

Budyko curves are used as a first order estimate of annual evaporation as a function of annual precipitation and potential evaporation. If the available energy is sufficient to evaporate the available moisture, annual evaporation can approach annual precipitation (water-limited situation). If the available energy is not sufficient, annual evaporation can approach potential evaporation (energy-limited situation). Using the water balance and the energy balance and by applying the definition of the aridity index and Bowen ratio, the Budyko framework can be described as (Arora, 2002):

$$\frac{E_a}{P_a} = \frac{\emptyset}{1+f(\emptyset)} = F(\emptyset)$$ (1)

with $E_a$ annual evaporation [L/T], $P_a$ annual precipitation [L/T], $\frac{E_a}{P_a}$ the evaporation ratio [-], and $\emptyset$ the aridity index which is defined as the potential evaporation divided by annual precipitation [-]. Equation 1 is the physical base of all Budyko curves, which are developed by different researchers (Table 1).

The equations shown in Table 1 assume that the evaporation ratio is determined by climate only and do not take into account the effect of other controls on the water balance. Therefor some researchers tried to incorporate more physics into the Budyko framework. For example Milly (1994, 1993) investigated the root zone storage as an important secondary control on the water balance. Choudhury (1999) used net radiation and a calibration factor in Budyko curves. Zhang et al. (2004, 2001) tried to add a plant-available water coefficient, Porporato et al. (2004) took into account the maximum storage capacity, and Yang et al. (2006, 2008) incorporated a catchment parameter, and Donohue et al. (2007) tried to consider vegetation dynamics. Although the incorporation of these additional processes improves the model performance, the main difficulty with these approaches is the determination of the parameter values. In practice, they are therefor often used as calibration parameters. The model of Gerrits et al. (2009) (hereafter Gerrits' model) aimed to develop an analytical model that is physically based and only uses measurable parameters. They tested the model output (i.e., interception evaporation, transpiration, and total evaporation) on a couple locations in the world, where the parameters could be determined, but not at the global scale due to data limitations. However, with the current developments in remotely sensed data new opportunities arise.

Recently, many studies (e.g., Chen et al., 2013; Donohue et al., 2010; Istanbulluoglu et al., 2012; Milly and Dunne, 2002; Wang, 2012; Zhang et al., 2008) found that soil water storage changes is a critical component in modelling of the interannual water balance. Including soil water information into the Budyko framework was often difficult, because this information is not widely available. However, in Gao et al. (2014) a new method is presented where the available soil water is derived from time series of rainfall and potential evaporation, plus a long-term runoff coefficient. This data can be derived locally (e.g., de Boer-Euser et al. (2016)), but can also be derived from remotely sensed data as shown by Wang-Erlandsson et al. (2016). While Gerrits' model was tested for 10 locations with different climate condition, the aim of this study is to test Gerrits' model at the global scale. Furthermore, we used the remotely sensed data for the parameters, which were considered constant in Gerrits' model. The remotely sensed data includes the estimation of the maximum soil moisture storage by the method of Gao et al (2014) and the estimation of the





required interception storage capacity values. These parameters are required to make a first order
estimate of total evaporation, and to partition this into interception evaporation and transpiration
as well. The outcome will be compared to more complex land-surface-atmosphere models as well
as to Budyko curves from Table 1.
**Methodology**
Total evaporation ($E$) may be partitioned as follows (Shuttleworth, 1993):

$$E = E_i + E_t + E_o + E_s \qquad (2)$$

in which $E_i$ is interception evaporation, $E_t$ is transpiration, $E_o$ is evaporation from water bodies
and $E_s$ is evaporation from the soil, all with dimensions [LT$^{-1}$]. In this definition, interception is
the amount of evaporation from any wet surface including canopy, floor, understory and the top
layer of the soil, which occurs on the same day as the rainfall. Soil evaporation is defined as
rainwater which is stored in the soil connected to the root zone (de Groen and Savenije, 2006) and
therefor is different from evaporation of the top layer of the soil (several millimeters of soil depth).
Gerrits et al. (2009) assumed that evaporation from the deep soil is negligible or can be combined
with interception evaporation. Evaporation from water bodies is used for the inland water for
which the interception evaporation and transpiration is zero. In that case, we can show equation 2
as follow:

$$E = E_o \qquad\qquad \text{for water bodies} \qquad (3)$$

$$E = E_i + E_t \qquad\qquad \text{others}$$

For modelling evaporation, it is important to consider that interception and transpiration have
different time scale (i.e. dividing the stock by the evaporative flux). With the stock amount of few
millimetres and the evaporative flux of a few millimetres per day, interception has a time scale in
the order of one day(Dolman and Gregory, 1992; A. M. J. Gerrits et al., 2009; Gerrits et al., 2007;
Savenije, 2004; Scott et al., 1995). In the case of transpiration, the stock amount of ten to hundreds
of millimetres and the evaporative flux of a few millimetres per day (Baird and Wilby, 1999),
results in a time scale in the order of month(s) (Gerrits et al., 2009). In Gerrits' model it is
successively assumed that interception and transpiration can be modelled as threshold processes
at the daily and monthly time scale, respectively. Rainfall characteristics are successively used to
temporally upscale from daily to monthly, and from monthly to annual. A full description of the
derivation and assumptions can be found in Gerrits et al. (2009). Here, we only summarize the
relevant equations (Table 2) and not the complete derivation. Since we now test the model at the
global scale, we do show how we estimated the required model parameters and the inputs we used.
**Interception**
The Gerrits' model considers evaporation from interception as a threshold process at daily time
scale (Equation 4). Daily interception ($E_{i,d}$), then, is upscaled to monthly interception ($E_{i,m}$,
Equation 5) by considering the frequency distribution of the rainfall on a rain day ($\beta$-parameter)
and later on to annual interception ($E_{i,a}$, Equation 6) by considering the frequency distribution of





the rainfall on a rain month ($\kappa_m$-parameter) (see de Groen and Savenije (2006), Gerrits et al.
(2009)). A rain day is defined as a day with more than 0.1 mm day$^{-1}$ of rain and a rain month is a
month with more than 2 mm month$^{-1}$ of rain.
While Gerrits et al. (2009) assumed a constant interception threshold ($D_{i,d} = 5$ mm day$^{-1}$) for the
studied locations, we here use a global valid value based on remote sensing data. The interception
threshold ($D_{i,d}$) is either limited by the daily interception storage capacity $S_{max}$ (mm day$^{-1}$) or by
the daily potential evaporation ($E_{p,d} = E_{p,a}/365$). $E_{p,a}$ is the annual potential evaporation (mm
year$^{-1}$):

$$D_{i,d} = \min(S_{max}, E_{p,d}) \tag{15}$$

The daily interception storage capacity should be seen as the total interception storage within one
day, including the (partly) emptying and filling of the storage between events, thus $S_{max} = n \cdot$
$C_{max}$, where $C_{max}$ is the interception storage capacity. If we assume on average maximal one rain
event per day ($n = 1$ day$^{-1}$) (Gerrits et al., 2010), $S_{max}$ [LT$^{-1}$] will approach $C_{max}$ [L] as often
found in literature. Despite proposing modifications for storms which last more than one day
(Pearce and Rowe, 1981) and multiple storms per rain day (Mulder, 1985), the modification is
rarely necessary (Miralles et al., 2010).
For $n = 1$, the interception storage capacity can be estimated from Von Hoyningen-Huene (1981),
which is obtained for a series of crops (de Jong and Jetten, 2007):

$$S_{max} \approx C_{max} = 0.935 + 0.498\text{LAI} - 0.00575\text{LAI}^2 \tag{16}$$

LAI is leaf area index derived from remote sensing images. Since the storage capacity of the forest
floor is not directly related to LAI, it could be said that the 0.935 mm is sort of the storage capacity
of the forest floor.
**Transpiration**
Transpiration is considered as a threshold process at the monthly time scale ($E_{t,m}$ (mm month$^{-1}$),
Equation 10) and successively is upscaled to annual transpiration ($E_{t,a}$ (mm year$^{-1}$), Equation 11)
by considering the frequency distribution of the net monthly rainfall ($P_{n,m} = P_m - E_{i,m}$) expressed
with the parameter $\kappa_n$. To estimate the monthly and annual transpiration, two parameters $A$ and $B$
are required. $A$ is the initial soil moisture or carryover value (mm month$^{-1}$) and $B$ is described as
follow:

$$B = 1 - \gamma + \gamma\exp(-\frac{1}{\gamma}) \tag{17}$$

and dimensionless $\gamma$ is equal to:

$$\gamma = \frac{S_b}{D_{t,m}\Delta t_m} \tag{18}$$





Gerrits et al. (2009) assumed that the caryover value ($A$) is constant and estimated annual
transpiration considering $A = 0$, $A = 5$, $A = 15$ or $A = 20$ mm month[-1] depending on the location.
Also they considered $\gamma$ to be constant ($\gamma = 0.5$). In the current study, we estimated these two
parameter using the maximum root zone storage capacity ($S_{u,max}$). We calculated $\gamma$ by equation
18. In this equation, $\Delta t_m = 1$ month and $S_b$ is the moisture content below which transpiration is
restricted. $S_b$ can be assumed to be 50% to 80% of $S_{u,max}$ (de Groen, 2002; Shuttleworth, 1993).
In this study we assumed $S_b$ to be 50% of $S_{u,max}$ as this value is commonly used for many crops
(Allen et al., 1998). Furthermore, we assumed that $A$ can be estimated as $bS_{u,max}$ and in this study
we assumed $b = 0.1$. To estimate $A$ and $\gamma$, it is important to have a reliable database of $S_{u,max}$.
For this purpose, we used the global estimation of $S_{u,max}$ from Wang-Erlandsson et al. (2016)
(Fig. 1d). $S_{u,max}$ is derived from the method of mass balance using the satellite based precipitation
and evaporation (Wang-Erlandsson et al., 2016). Wang-Erlandsson et al. (2016) estimated the root
zone storage capacity from soil moisture deficit constructed from water outflow (i.e. evaporation
which is sum of transpiration, evaporation, interception, soil moisture evaporation and open water
evaporation) and inflow (i.e. precipitation and irrigation). In their study, the root zone storage
capacity is defined as the total plant available water including the deep rooting system of plants to
survive droughts. Note that this recent method (Gao et al., 2014) to estimate $S_{u,max}$ is not using
soil information, which is often used, but only uses climatic data. For arid climates the difference
between this method and the soil-derived methods are limited (de Boer-Euser et al., 2016).
Furthermore, Gerrits et al. (2009) estimated the monthly transpiration threshold ($D_{t,m}$) as $\frac{E_P - E_{i,a}}{n_a}$
which assumes that if there is little interception, plants can transpire at the same rate as a well-
watered reference grass as calculated with the Penman-Monteith equation (University of East
Anglia et al., 2014). In reality, most plants encounter more resistance (crop resistance) than grass,
hence we used the relation found by (Novák and Ján, 2005) to convert potential evaporation of
reference grass ($E_P$) to potential transpiration of certain crop depending on LAI (i.e. the
transpiration threshold $D_{t,m}$ [mm month[-1]]):

$$D_{t,m} = \frac{E_P}{n_a}(1 - \exp(-\beta \text{LAI})) \tag{19}$$

in which $E_P$ is annual potential evaporation (for open water) (mm year[-1]), $n_a$ is the number of
months in a year (=12), LAI is canopy leaf area index and $\beta$ is a coefficient between 0.45 and 0.55
and $\beta = 0.463$ is valid for a large number of agricultural canopies (Novák and Ján, 2005). Our
primary investigation also showed that $\beta = 0.463$ is also valid for other land cover types including
evergreen, deciduous and mixed forests.
**Data**
For precipitation we used the AgMERRA product from AgMIP climate forcing dataset (Ruane et
al., 2015), which has a daily time scale and a spatial resolution of 0.25°×0.25° (see Fig. 1a). The
spatial coverage of AgMERRA is globally for the years 1980-2010. The AgMERRA product is
available    on    the    NASA    Goddard    Institute    for    Space    Studies    website
(http://data.giss.nasa.gov/impacts/agmipcf/agmerra/).





Potential evaporation (see Fig. 1b) data (calculated by FAO-Penman–Monteith equation (Allen et
al., 1998)) were taken from Center for Environmental Data Archival website
(http://catalogue.ceda.ac.uk/uuid/4a6d071383976a5fb24b5b42e28cf28f), produced by the
Climatic Research Unit (CRU) at the University of East Anglia (University of East Anglia Climatic
Research Unit, 2014). These data are at the monthly time scale over the period 1901-2013, and has
a spatial resolution of 0.5°×0.5°. We used the data of 1980-2010 in consistent with precipitation
dataset.
LAI data (Fig. 1c) were obtained from Vegetation Remote Sensing & Climate Research
(http://sites.bu.edu/cliveg/datacodes/) (Zhu et al., 2013). The spatial resolution of the data sets is
1/12 degree, with 15-day composites (2 per month) for the period July 1981 to December 2011.
The data of $S_{u,max}$  (Fig. 1d) is prepared data by Wang-Erlandsson et al. (2016) and is based on
the satellite based precipitation and evaporation with 0.5°×0.5° resolution over the period 2003-
2013. They used the USGS Climate Hazards Group InfraRed Precipitation with Stations (CHIRPS)
precipitation data at 0.05° (Funk et al., 2014) and the ensemble mean of three datasets of
evaporation including CSIRO MODIS Reflectance Scaling EvapoTranspiration (CMRSET) at
0.05° (Guerschman et al., 2009), the Operational Simplified Surface Energy Balance (SSEBop) at
30″ (Senay et al., 2013) and MODIS evapotranspiration (MOD16) at 0.05° (Mu et al., 2011). They
calculated potential evaporation using Penman-Monteith equation (Monteith, 1965).
**Model comparison and evaluation**
The model performance was evaluated by comparing our results at the global scale to global
evaporation estimates from other studies. Most available products only provide total evaporation
estimates and do not distinguish between interception and transpiration. Therefore, we chose to
compare our interception and transpiration results to two land surface models: The Global Land
Evaporation Amsterdam Model (GLEAM) (v3.0a) database (Miralles et al., 2011, Martens et al.
2016) and Simple Terrestrial Evaporation to Atmosphere Model (STEAM) (Wang-Erlandsson et
al., 2014, Wang-Erlandsson et al., 2016). GLEAM estimates different fluxes of evaporation
including transpiration, interception, bare soil evaporation, snow sublimation and open water
evaporation. STEAM, on the other hand, estimates the different components of evaporation
including transpiration, vegetation interception, floor interception, soil moisture evaporation, and
open water evaporation. Thus for the comparison of interception we used the sum of canopy and
floor interception and soil evaporation from STEAM and canopy interception and bare soil
evaporation from GLEAM. Furthermore, STEAM includes an irrigation module (Wang-
Erlandsson et al., 2014), while Miralles et al. (2011) mentioned that they did not include irrigation
in GLEAM, but the assimilation of the soil moisture from satellite would account for it as soil
moisture adjusted to seasonal dynamics of any region. The total evaporation was also compared to
LandFlux-EVAL products (Mueller et al., 2013). GLEAM database (www.gleam.eu) is available
for 1980-2014 with a resolution of 0.25°×0.25° and STEAM model was performed for 2003-2013
with a resolution of 1.5°×1.5°. LandFlux-EVAL data (https://data.iac.ethz.ch/landflux/) is
available for 1989-2005. We compared Gerrits' model to other products based on the land cover
to judge the performance of the model for different types of land cover. The global land cover map
(Channan et al., 2014; Friedl et al., 2010) was obtained from http://glcf.umd.edu/data/lc/. Lastly,
we also compared our results to the Budyko curves of Schreiber, O'ldekop, Pike and Budyko





(Table 1). We used coefficient of determination ($R^2$), root mean square error (*RMSE*) (Eq. 20),
mean bias error (*MBE*) (Eq. 21) and relative error (*RE*) (Eq. 22) to evaluate the results.

$$\text{RMSE} = \sqrt{\frac{\sum_{i=1}^{n}(x_{iG} - x_{iM})^2}{n}} \tag{20}$$

$$\text{MBE} = \frac{\sum_{i=1}^{n}(x_{iG} - x_{iM})}{n} \tag{21}$$

$$\text{RE} = \frac{\bar{x}_G - \bar{x}_M}{\bar{x}_G} \times 100 \tag{22}$$

In these equations, $x_{iM}$ is evaporation of the benchmark models to which Gerrits' model is
compared for pixel $i$, $x_{iG}$ is evaporation from Gerrits' model for pixel $i$, $\bar{x}_G$ is the average
evaporation of Gerrits' model, $\bar{x}_M$ is the average evaporation of the benchmark models and $n$ is
the number of pixels of the evaporation map. Negative MBE and RE show the Gerrits' model
underestimates evaporation and vice versa. As the spatial resolution of the products is different,
we regridded all the products to the coarsest resolution (1.5°×1.5°) for the comparison.
**Results and discussion**
**Total evaporation comparison**
Figure 2 shows the mean annual evaporation from Gerrits' model, Landflux-EVAL, STEAM and
GLEAM data sets. In general, the spatial distribution of Gerrits' simulated interception is partly
similar to that of the benchmark models. Figure 2a demonstrates that, as expected, the highest
annual evaporation, which is the sum of interception evaporation and transpiration, occurs in
tropics with evergreen broadleaf forests and the lowest rate occurs in the barren and sparsely
vegetated regions like north of Africa, Saudi Arabia, parts of Iran, China, Turkmenistan,
Uzbekistan, Kazakhstan and Chile. Total evaporation varies between almost zero in arid regions
and more than 1500 mm year$^{-1}$ in the tropics. The differences can be seen in the central Africa and
in the arid and semi-arid aria such as Saudi Arabia, parts of Iran, China, Turkmenistan, Uzbekistan,
Kazakhstan and Gobi Desert.
Mean annual evaporation contributions per land cover type from Gerrits' model and other products
as well as RMSE, MBE and RE are shown in Table 3. Globally, mean annual evaporation
estimated by Gerrits' model, Landflux-EVAL, STEAM and GLEAM is 515, 511, 511 and 511
mm year$^{-1}$, respectively. The highest mean annual evaporation rates are found in Evergreen
broadleaf forests, Savannas and Deciduous broadleaf forests. The lowest values of mean annual
evaporation are found in Shrublands, Grasslands and Deciduous needleleaf forests. Generally,
Gerrits' model overestimates evaporation for most land cover types in comparison to Landflux-
EVAL and GLEAM, and underestimates in comparison to STEAM (see also MBE and RE).
RMSE, MBE and RE for each land cover type show that, generally, Gerrits' model is in a better
agreement with Landflux and GLEAM than STEAM. The scatter plot of total evaporation
estimated by Gerrits' model in comparison to Landflux-EVAL, STEAM and GLEAM for each
land cover type (Fig. 3) also indicates that Gerrits' model has a better agreement with Landflux-
EVAL and GLEAM than STEAM model, especially for Evergreen broadleaf forest, Shrublands,





Savannas and Croplands. Since the number of pixels covered by each land use is different, RMSE,
MBE and RE can not be comparable between land cover types.
It should be mentioned that we intercompared all products as well and found that, in general, there
are also big differences between STEAM, GLEAM and Landflux-EVAL. Different products of
precipitation (and other global data bases) applied for the models can be a convincing reason. For
example, the sensitivity of the model to the number of rain days and rain months especially for the
higher rate of precipitation (Gerrits et al., 2009) can be a probable reason for poor performance of
the model especially for evergreen forests with the higher amount of precipitation.
**Annual interception comparison**
While Wang-Erlandsson et al. (2014) estimated the canopy interception, floor interception and soil
evaporation separately, in the current study we assumed that these three components of
evaporation can be estimated together by equation 16 as interception evaporation. Figure 4 shows
the mean annual evaporation from interception at the global scale estimated by Gerrits' model,
STEAM and GLEAM. It should be mentioned that in this figure, interception from STEAM is
calculated by the sum of canopy interception, floor interception and soil evaporation. Furthermore,
interception from GLEAM is calculated as the sum of canopy interception and bare soil
evaporation (GLEAM does not estimate floor interception). In general, the spatial distribution of
Gerrits' simulated interception is partly similar to that of STEAM and GLEAM. In the tropics,
with high amount of annual precipitation and high storage capacity due to the dense vegetation
(evergreen broadleaf forests and savannas), annual interception shows the highest values. Table 4
shows the average of interception, RMSE, MBE and RE per land cover type. This table indicates
that Gerrits' model underestimates interception in comparison to STEAM for all land cover types
except for Savannas (MBE=+10 mm year$^{-1}$) and Croplands (MBE=+8 mm year$^{-1}$). Table 4 also
shows that, in comparison to GLEAM, Gerrits' model overestimates interception for all land cover
types, because in GLEAM floor interception has not been taken into account. Figure 5 also shows
that Gerrits' model is in reasonable agreement with STEAM (especially for Grasslands and Mixed
forest) rather than GLEAM. The reason for the overestimated interception could be the role of the
understory. LAI does not account for understory, therefore maybe $S_{max}$ should be larger than
modeled with equation 16. However, there is almost no data available to estimate the interception
storage capacity of the forest floor at the global scale. Although, on the other hand, it could be said
that the 0.935 mm in equation 16 is the forest floor interception storage capacity.
**Annual transpiration comparison**
Figure 6 illustrates the mean annual transpiration estimated by Gerrits' model, STEAM and
GLEAM. The spatial distribution is partly similar to the results of STEAM and GLEAM. Mean
annual transpiration varies between zero mm year$^{-1}$ for arid areas in the north of Africa (Sahara) to
more than 1000 mm year$^{-1}$ in the tropics in south America. The results show that the highest annual
transpiration occurrs in Evergreen broadleaf forests with the highest amount of precipitation and
dense vegetation (see also Table 5). Figure 6c shows that GLEAM, in comparison to Gerrits'
model, overestimates the transpiration in some regions especially in the tropics in south America
and central Africa. Figure 6b also shows that STEAM is different from Gerrits' model over some
regions like India, west of China and North America as well as tropics. Table 5 (MBE and RE)





also indicates that Gerrits' model underestimates transpiration in comparison to GLEAM and
overestimates in comparison STEAM. In Gerrits' model, we neglected the effect of seasonality on
the transpiration threshold. Since most vegetation species have a dormant period, this assumption
causes an error in Gerrits' model. The scatter plot of transpiration (Fig. 7) also shows that Gerrits'
model underestimates transpiration in comparison to GLEAM and overestimates in comparison to
STEAM. All land cover types show a reasonable agreement between Gerrits' model and other
products.
**Budyko framework**
Figure 8 shows the mean annual evaporation derived from four non-parametric Budyko curves
(Table 1) including Schreiber (1904), Ol'dekop (1911), Pike (1964) and Budyko (1974). The
global mean annual evaporation estimated by Pike and Budyko are similar and evaporation from
Budyko is geometric mean of Schreiber and Ol'dekop curves. Schreiber underestimates the mean
annual evaporation in comparison to Ol'dekop, Pike and Budyko, especially in regions with a
higher rate of evaporation. Table 6 shows the mean annual evaporation estimated by these four
curves per land cover type in comparison to Gerrits' model as well as RMSE, MBE and RE. The
results show that evaporation of Gerrits' model is closer to that of Ol'dekop, especially for
Deciduous broadleaf forest and Shrublands (see also Fig. 9). The scatter plot of evaporation (Fig.
9) also shows that, in comparison to Budyko curves, Gerrits' model performs well for all land
cover types except for Evergreen broadleaf and Deciduous needleleaf forest. As can be seen, the
difference between Ol'dekop and Gerrits' model is less than the others and for most parts of the
world. Evergreen broadleaf forest shows a significant overestimation of evaporation by Gerrits'
model in comparison to Budyko curves. One of the reason for these differences can be the used
precipitation product as Gerrits et al. (2009) mentioned that the number of rain months per year,
is the most sensitive parameter. Furthermore, as mentioned before ("Annual interception
comparison" section), the role of understory, which has not been taken into account in $S_{max}$
equation, can be a source of error for the poor interception performance (and therefore total
evaporation) in forests.
**Conclusion**
In the current study we improved and applied a simple evaporation model proposed by Gerrits et
al. (2009) at the global scale. Instead of locally determined model parameters we now only used
parameters derived from remotely sensed data. Furthermore, we implemented in the Gerrits' model
a new definition of the available soil water from Gao et al (2014).
The spatial distribution of evaporation shows that the highest annual evaporation occurs in tropics
with Evergreen broadleaf forests and the lowest rate occurs in the barren and sparsely vegetated
regions. Total evaporation varies between almost zero in arid regions and more than 1500 mm
year$^{-1}$ in the tropics. The spatial distribution of evaporation of Gerrits' model is in good agreement
with STEAM, GLEAM, and Landflux-EVAL.
Comparing our results for total evaporation to Landflux-EVAL estimates shows that Gerrits'
model is in good agreement with Landflux-EVAL. The highest mean annual evaporation rates are
found in evergreen broadleaf forests (1286 mm year$^{-1}$), deciduous broadleaf forests (733 mm year$^{-}$



$^1$) and savannas (721 mm year$^{-1}$) and the lowest ones are found in shrublands (254 mm year$^{-1}$) and
grasslands (305 mm year$^{-1}$). Generally, Gerrits' model overestimates in comparison to Landflux-
EVAL and GLEAM, and underestimates in comparison to STEAM.
Gerrits' model underestimates interception in comparison to STEAM for all land covers excluding
savannas (MBE=+10 mm year$^{-1}$) and croplands (MBE=+8 mm year$^{-1}$). On the other hand, the
model overestimates interception in comparison to GLEAM, since GLEAM does not include floor
interception. Although we tried to correct for the different definitions of interception, the results
may be biased hereby. The relatively worse performance in forests ecosystems could be explained
by the effect of understory. This is not taken into account in Gerrits' model, while the understory
can also intercept water. Although we could say that the constant value of 0.935 mm in equation
16 is the forest floor interception storage capacity. Therefore, better estimation of $S_{max}$ to better
estimate forest floor interception is recommended.
Estimated transpiration by Gerrits' model is in reasonable agreement with GLEAM and STEAM.
Gerrits' model underestimates transpiration in comparison to GLEAM (RE=-10%) and
overestimates in comparison to STEAM (RE=+12%). The scatter plots showed that, in comparison
to GLEAM and STEAM, Gerrits' model perform well for all land cover types.
Comparing Gerrits' model to some Budyko curves, shows that the model performed well, but in
areas with few number of rain months, evaporation is not close to the Budyko curves of Schreiber,
Ol'dekop, Pike and Budyko. This is likely caused by the fact that Gerrits' model is rather sensitive
to the number of rain days and months.
Gerrits' model partitioned evaporation into interception and transpiration, while GLEAM and
STEAM partitioned evaporation into more components such as soil moisture and so on. Therefore,
it is a source of error if we compare interception and transpiration of Gerrits' model to those of
GLEAM and STEAM.
As we compared all products together, we also found that, in general, there are also big differences
between STEAM, GLEAM and Landflux-EVAL. The most convincing reason of this discrepancy
can be the different products of precipitation (and other global data bases), which is used for the
models. The Gerrits' model is sensitive to the number of rain days and months especially for the
higher rate of precipitation. Therefore, fore evergreen forest with the higher amount of
precipitation, this issue can be a probable reason. But it should be mentioned that the strong point
of the Gerrits' model is that, in comparison to other models, it is a very simple model and in spite
of the simplicity, the Gerrits' model performs quite well.
**Acknowledgment**
This research was partly funded by NWO Earth and Life Sciences (ALW), veni-project
863.15.022, the Netherlands. Furthermore, we would like to thank Iran's Ministry of Science,
Research and Technology for supporting this research and the mobility fellowship. We also would
like to thank Jie Zhou, Lan Wang-Erlandsson, Kamran Davary, Shervan Gharari and Hubert
Savenije for their kind helps and comments.





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





1    **Table 1-** Budyko equations developed by different researchers.

| Equation | Reference |
|---|---|
| $\dfrac{E_a}{P_a} = 1\text{-exp}(-\emptyset)$ | Schreiber [1904] |
| $\dfrac{E_a}{P_a} = \emptyset\tanh(\dfrac{1}{\emptyset})$ | Ol'dekop [1911] |
| $\dfrac{E_a}{P_a} = \dfrac{1}{\sqrt{0.9 + (\dfrac{1}{\emptyset})^2}}$ | Turc [1954] |
| $\dfrac{E_a}{P_a} = \dfrac{1}{\sqrt{1 + (\dfrac{1}{\emptyset})^2}}$ | Pike [1964] |
| $\dfrac{E_a}{P_a} = [\emptyset \tanh\left(\dfrac{1}{\emptyset}\right)(1\text{-exp}(-\emptyset))]^{1/2}$ | Budyko [1974] |




**Table 2-** Summary of the interception and transpiration equations of Gerrits' model at different
time scales (Gerrits et al., 2009) ($E_{i,d}$: daily interception (mm day$^{-1}$), $P_d$: daily precipitation (mm
day$^{-1}$), $D_{i,d}$: the daily interception threshold (mm day$^{-1}$), $E_{i,m}$: monthly interception (mm month$^{-1}$),
$P_m$: monthly rainfall (mm month$^{-1}$), $\emptyset_{i,m}$: a sort of aridity index for interception at monthly scale,
$E_{i,a}$: annual interception (mm year$^{-1}$), $P_a$: annual rainfall (mm year$^{-1}$), $\emptyset_{i,a}$: a sort of aridity index
for interception at annual scale, $K_0$ and $K_1$: the Bessel function of the first and second order,
respectively, $E_{t,m}$ monthly transpiration (mm month$^{-1}$), A: carry-over parameter (mm month$^{-1}$),
$D_{t,m}$: the transpiration threshold (mm month$^{-1}$), $E_{t,a}$: annual transpiration (mm year$^{-1}$), $\emptyset_{t,a}$: an
aridity index and B: slope of relation between monthly effective rainfall and monthly transpiration.

| Time scale | Interception | | Transpiration | |
|---|---|---|---|---|
| Daily | $E_{i,d} = \min(D_{i,d}, P_d)$ | (4) | - | |
| Monthly | $E_{i,m} = P_m\left(1 - \exp(-\emptyset_{i,m})\right)$ | (5) | $E_{t,m} = \min(A + B(P_m - E_{i,m}), D_{t,m})$ | (10) |
| Annual | $E_{i,a} = P_a(1 - 2\emptyset_{ia}K_0(2\sqrt{\emptyset_{i,a}}) - 2\sqrt{\emptyset_{i,a}}K_1(2\sqrt{\emptyset_{i,a}}))$ | (6) | $E_{t,a} = 2BP_a\left(\emptyset_{ia}K_0(2\sqrt{\emptyset_{i,a}}) + \sqrt{\emptyset_{i,a}}K_1(2\sqrt{\emptyset_{i,a}})\right)\left(\frac{A}{\kappa_n B} + 1 - \exp(-\emptyset_{t,a})\left(\frac{A}{\kappa_n B} + 1 + \emptyset_{t,a} - \frac{\emptyset_{t,a}}{B}\right)\right)$ | (11) |
| with | $\emptyset_{i,m} = \dfrac{D_{i,d}}{\beta}$ | (7) | $\beta = \dfrac{P_m}{E(n_{r,d}\vert n_m)}$ | (12) |
| | $\emptyset_{i,a} = \dfrac{n_{r,d}D_{i,d}}{\kappa_m}$ | (8) | $\kappa_m = \dfrac{P_a}{E(n_{r,m}\vert n_a)} \approx \dfrac{P_a}{n_m}$ | (13) |
| | $\emptyset_{t,a} = \dfrac{D_{t,m}}{\kappa_n}$ | (9) | $\kappa_n = \dfrac{P_{n,a}}{E(n_{nr,m}\vert n_a)} = \dfrac{P_a - E_{i,a}}{E(n_{nr,m}\vert n_a)}$ | (14) |



**Table 3-** Comparison of mean annual evaporation estimated by Gerrits' model to Landflux-EVAL, STEAM and GLEAM through Average, RMSE, MBE and RE per land cover type. Negative MBE and RE show the Gerrits' model underestimates evaporation and vice versa. Average, RMSE and MBE are in mm year⁻¹ and RE is in %.

| Land cover | area | Gerrits | Landflux-EVAL | | | | STEAM | | | | GLEAM | | | |
|---|---|---|---|---|---|---|---|---|---|---|---|---|---|---|
| | 1000km² | Avg.* | Avg. | RMSE | MBE | RE | Avg. | RMSE | MBE | RE | Avg. | RMSE | MBE | RE |
| Evergreen needleleaf forest | 5563 | 444 | 398 | 121 | +46 | +10 | 458 | 134 | -14 | -3 | 480 | 127 | -36 | -8 |
| Evergreen broadleaf forest | 11778 | 1286 | 1202 | 209 | +84 | +7 | 1178 | 296 | +108 | +8 | 1260 | 203 | +26 | +2 |
| Deciduous needleleaf forest | 2498 | 325 | 288 | 60 | +37 | +11 | 348 | 55 | -23 | -7 | 337 | 55 | -12 | -4 |
| Deciduous broadleaf forest | 1106 | 733 | 736 | 117 | -3 | -0.4 | 824 | 178 | -91 | -12 | 662 | 132 | +71 | +10 |
| Mixed forest | 13470 | 505 | 473 | 125 | +32 | +6 | 521 | 155 | -17 | -3 | 515 | 136 | -10 | -2 |
| Shrublands¹ | 29542 | 254 | 249 | 68 | +5 | +2 | 228 | 96 | +26 | +10 | 249 | 83 | +5 | +2 |
| Savannas² | 18846 | 721 | 766 | 120 | -45 | -6 | 756 | 189 | -35 | -5 | 722 | 124 | -0.8 | -0.1 |
| Grasslands | 21844 | 305 | 343 | 91 | -37 | -12 | 325 | 132 | -20 | -7 | 332 | 114 | -27 | -9 |
| Croplands | 12417 | 547 | 535 | 105 | +12 | +2 | 557 | 186 | -10 | -2 | 489 | 119 | +58 | +11 |
| Croplands and natural vegetation mosaic | 5782 | 676 | 727 | 169 | -50 | -7 | 734 | 271 | -58 | -9 | 646 | 156 | +30 | +4 |
| Total (all land classes) | 122846 | 515 | 511 | 116 | +4 | +0.9 | 511 | 169 | +4 | +0.8 | 511 | 124 | +4 | +0.8 |

¹including open and closed shrublands. ²including woody savannas and savannas.



**Table 4-** Comparison of interception estimated by Gerrits' model to STEAM and GLEAM through Average, RMSE, MBE and RE per land cover type. Negative MBE and RE show the Gerrits' model underestimates evaporation and vice versa. Average, RMSE and MBE are in mm year$^{-1}$ and RE is in %.

| Land cover | Area | Gerrits | STEAM | | | | GLEAM | | | |
|---|---|---|---|---|---|---|---|---|---|---|
| | 1000km² | Avg. | Avg. | RMSE | MBE | RE | Avg. | RMSE | MBE | RE |
| Evergreen needleleaf forest | 5563 | 154 | 209 | 70 | -55 | -36 | 144 | 65 | +10 | +7 |
| Evergreen broadleaf forest | 11778 | 504 | 511 | 135 | -6 | -1 | 349 | 180 | +155 | +31 |
| Deciduous needleleaf forest | 2498 | 104 | 163 | 62 | -59 | -57 | 25 | 81 | +79 | +76 |
| Deciduous broadleaf forest | 1106 | 256 | 307 | 79 | -51 | -20 | 73 | 187 | +183 | +72 |
| Mixed forest | 13470 | 180 | 210 | 59 | -30 | -17 | 124 | 78 | +55 | +31 |
| Shrublands[1] | 29542 | 82 | 112 | 44 | -30 | -36 | 61 | 57 | +21 | +26 |
| Savannas[2] | 18846 | 257 | 247 | 85 | +10 | +4 | 106 | 172 | +150 | +59 |
| Grasslands | 21844 | 114 | 135 | 49 | -21 | -19 | 92 | 72 | +21 | +19 |
| Croplands | 12417 | 183 | 174 | 66 | +8 | +4 | 97 | 102 | +85 | +47 |
| Croplands and natural vegetation mosaic | 5782 | 216 | 247 | 112 | -31 | -15 | 104 | 145 | +112 | +52 |
| Total (all land classes) | 122846 | 184 | 203 | 73 | -21 | -10 | 116 | 102 | +58 | +37 |

[1]including open and closed shrublands. [2]including woody savannas and savannas.





**Table 5-** Comparison of transpiration estimated by Gerrits' model to STEAM and GLEAM through Average, RMSE, MBE and RE per land cover type. Negative MBE and RE show the Gerrits' model underestimates evaporation and vice versa. Average, RMSE and MBE are in mm year⁻¹ and RE is in %.

| Land cover | Area | Gerrits | STEAM | | | | GLEAM | | | |
|---|---|---|---|---|---|---|---|---|---|---|
| | 1000km² | Avg. | Avg. | RMSE | MBE | RE | Avg. | RMSE | MBE | RE |
| Evergreen needleleaf forest | 5563 | 290 | 209 | 123 | +81 | +28 | 258 | 115 | +32 | +11 |
| Evergreen broadleaf forest | 11778 | 781 | 659 | 209 | +123 | +16 | 897 | 182 | -115 | -15 |
| Deciduous needleleaf forest | 2498 | 221 | 182 | 56 | +39 | +18 | 260 | 65 | -39 | -18 |
| Deciduous broadleaf forest | 1106 | 477 | 499 | 114 | -22 | -5 | 578 | 142 | -101 | -21 |
| Mixed forest | 13470 | 325 | 288 | 121 | +37 | +11 | 352 | 110 | -27 | -8 |
| Shrublands[1] | 29542 | 172 | 108 | 92 | +65 | +38 | 156 | 67 | +17 | +10 |
| Savannas[2] | 18846 | 464 | 485 | 133 | -21 | -5 | 597 | 180 | -133 | -29 |
| Grasslands | 21844 | 191 | 175 | 94 | +16 | +9 | 198 | 139 | -7 | -4 |
| Croplands | 12417 | 364 | 359 | 116 | +5 | +1 | 377 | 93 | -13 | -3 |
| Croplands and natural vegetation mosaic | 5782 | 461 | 455 | 187 | +5 | +1 | 522 | 149 | -62 | -13 |
| Total (all land classes) | 122846 | 331 | 291 | 124 | +39 | +12 | 364 | 126 | -33 | -10 |

[1]including open and closed shrublands. [2]including woody savannas and savannas.

**Table 6-** Comparison of mean annual evaporation estimated by Gerrits' model to Schreiber, Ol'dekop, Pike and Budyko through Average, RMSE, MBE and RE per land cover type. Negative MBE and RE show the Gerrits' model underestimates evaporation and vice versa. Average, RMSE and MBE are in mm year⁻¹ and RE is in %.

| Land cover | area 1000km² | Gerrits Avg. | Schreiber Avg. | RMSE | MBE | RE | Ol'dekop Avg. | RMSE | MBE | RE | Pike Avg. | RMSE | MBE | RE | Budyko Avg. | RMSE | MBE | RE |
|---|---|---|---|---|---|---|---|---|---|---|---|---|---|---|---|---|---|---|
| Evergreen needleleaf forest[1] | 5563 | 444 | 358 | 132 | +86 | +19 | 428 | 97 | +16 | +4 | 399 | 106 | +45 | +10 | 391 | 110 | +53 | +12 |
| Evergreen broadleaf forest | 11778 | 1286 | 879 | 437 | +406 | +32 | 1070 | 269 | +216 | +17 | 996 | 330 | +290 | +23 | 970 | 354 | +315 | +25 |
| Deciduous needleleaf forest | 2498 | 325 | 249 | 89 | +76 | +23 | 288 | 58 | +37 | +11 | 272 | 70 | +54 | +16 | 268 | 73 | +57 | +18 |
| Deciduous broadleaf forest | 1106 | 733 | 654 | 120 | +79 | +11 | 740 | 52 | -7 | -0.9 | 701 | 72 | +31 | +4 | 695 | 80 | +38 | +5 |
| Mixed forest | 13470 | 505 | 403 | 160 | +102 | +20 | 486 | 124 | +19 | +4 | 451 | 133 | +53 | +11 | 443 | 137 | +62 | +12 |
| Shrublands[1] | 29542 | 254 | 240 | 56 | +14 | +6 | 263 | 48 | -9 | -3 | 253 | 49 | +2 | +0.7 | 251 | 50 | +3 | +1 |
| Savannas[2] | 18846 | 721 | 657 | 122 | +63 | +9 | 764 | 102 | -43 | -6 | 718 | 91 | +3 | +0.4 | 708 | 94 | +12 | +2 |
| Grasslands | 21844 | 305 | 324 | 83 | -19 | -6 | 344 | 89 | -39 | -13 | 333 | 84 | -28 | -9 | 334 | 84 | -29 | -9 |
| Croplands | 12417 | 547 | 519 | 115 | +28 | +5 | 584 | 115 | -37 | -7 | 555 | 107 | -8 | -1 | 550 | 107 | -3 | -0.6 |
| Croplands and natural vegetation mosaic | 5782 | 676 | 634 | 174 | +43 | +6 | 725 | 156 | -49 | -7 | 685 | 154 | -9 | -1 | 678 | 157 | -1 | -0.2 |
| Total (all land classes) | 122846 | 515 | 442 | 172 | +73 | +14 | 509 | 122 | +6 | +1 | 481 | 136 | +35 | +7 | 474 | 143 | +41 | +8 |

[1]including open and closed shrublands. [2]including woody savannas and savannas.





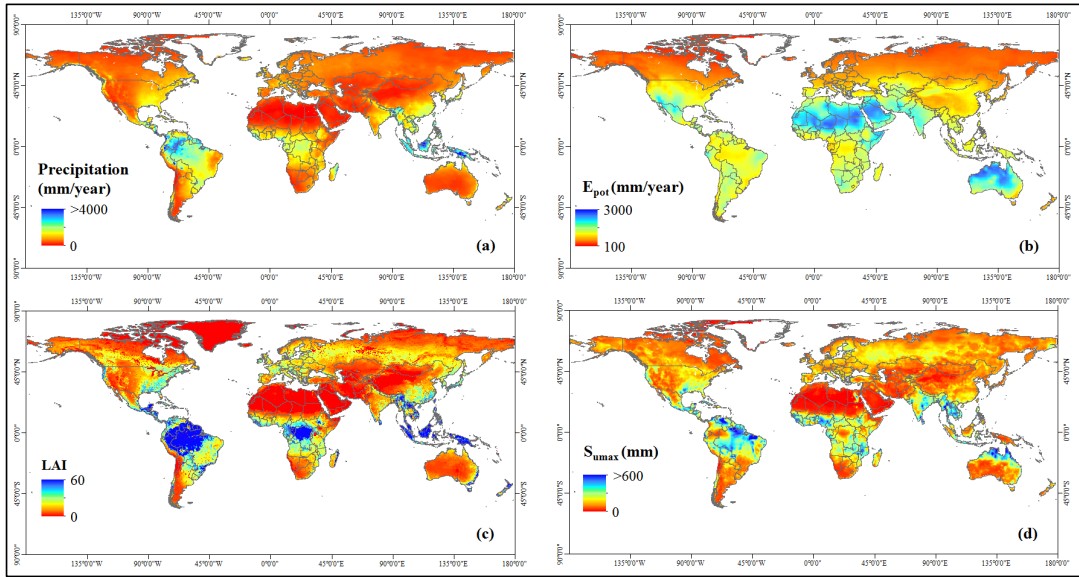

**Figure 1-** Mean annual of the applied data in the current study: (a) Precipitation (Ruane et al., 2015), (b) Potential evaporation (University of East Anglia Climatic Research Unit, 2014), (c) LAI (Zhu et al., 2013) and (d) $S_{u,max}$ (Wang-erlandsson et al., 2016).



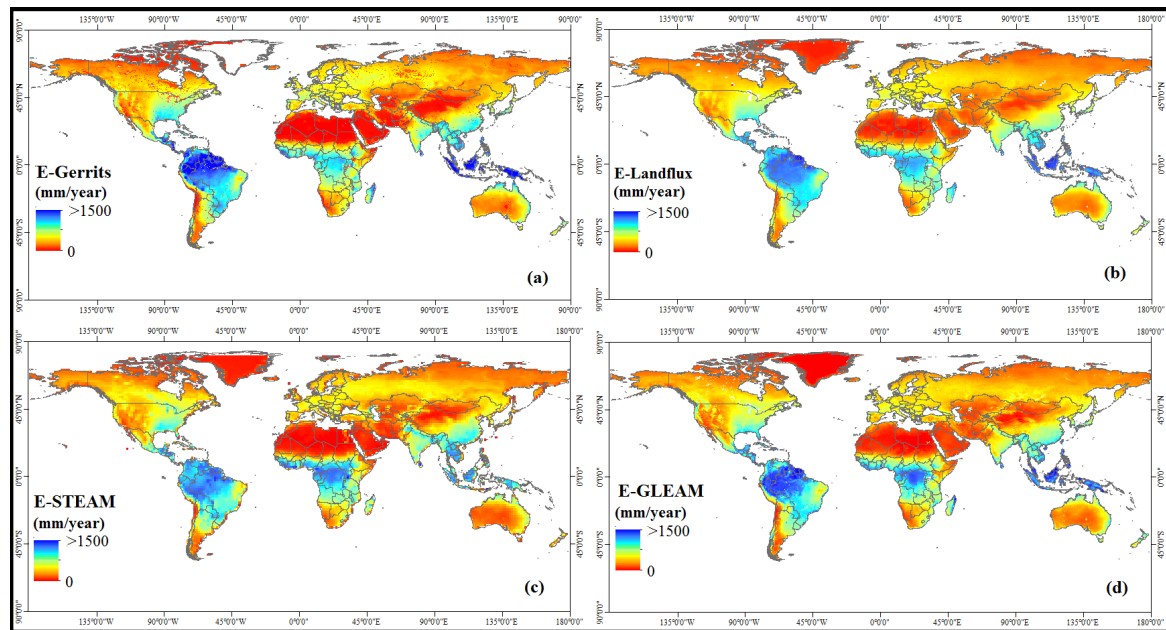

**Figure 2-** Mean annual evaporation estimated by (a) Gerrits' model, (b) Landflux-EVAL, (c)
STEAM and (d) GLEAM.





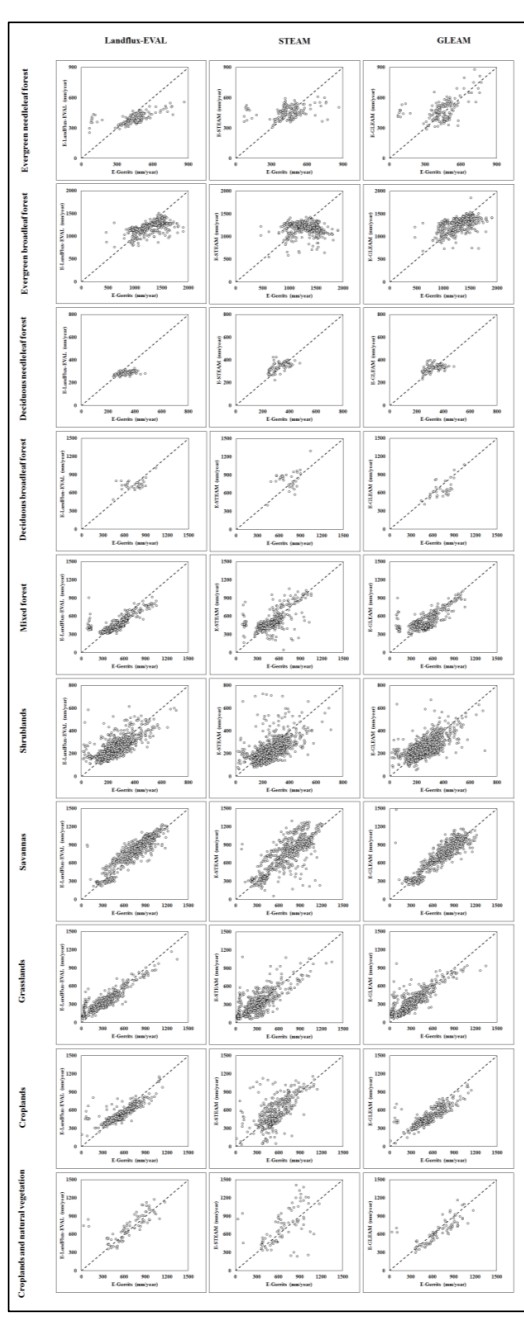

**Figure 3-** Scatter plot of mean annual evaporation estimated by Gerrits' model in comparison to Landflux-EVAL (left panel), STEAM (middle panel) and GLEAM (right panel) per land cover type.



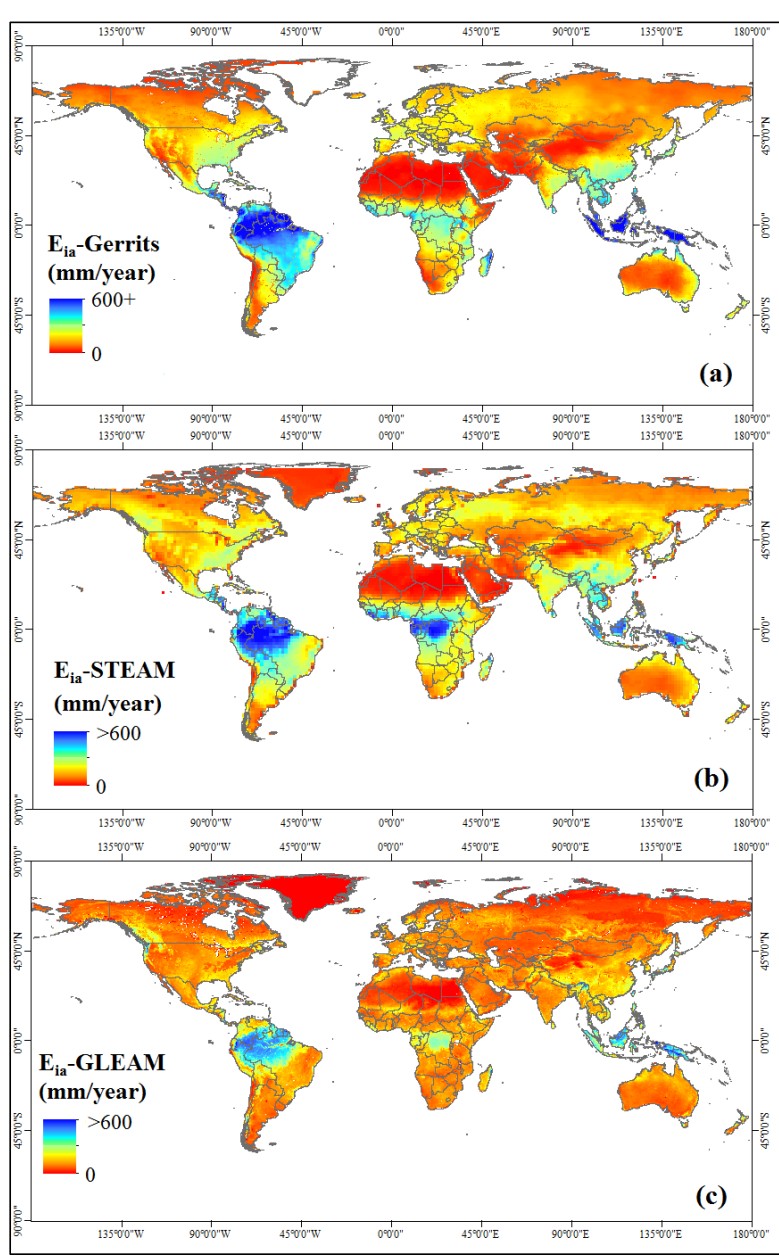



**Figure 4-** Simulated mean annual interception by (a) Gerrits' model and (b) STEAM and (c) GLEAM.





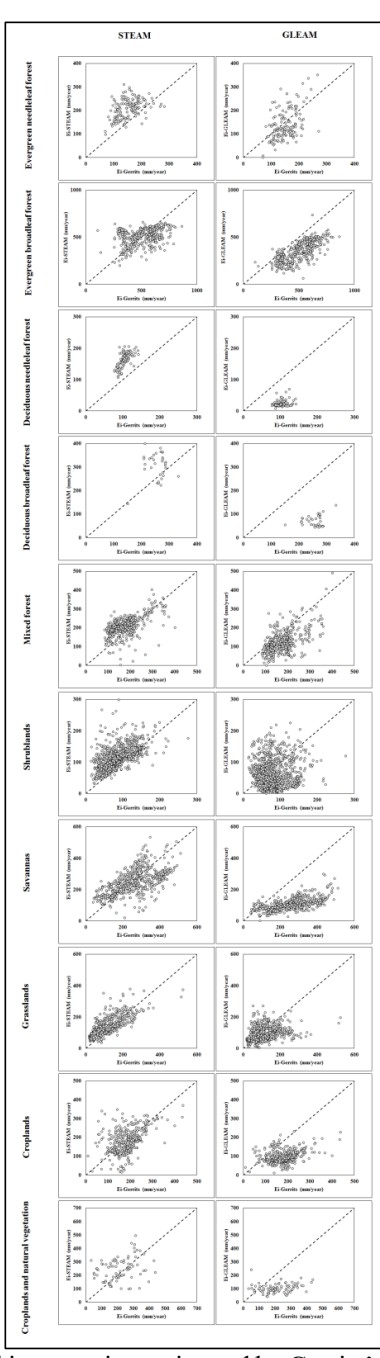

2 **Figure 5-** Scatter plot of annual interception estimated by Gerrits' model in comparison to STEAM
3 and GLEAM per land cover type.





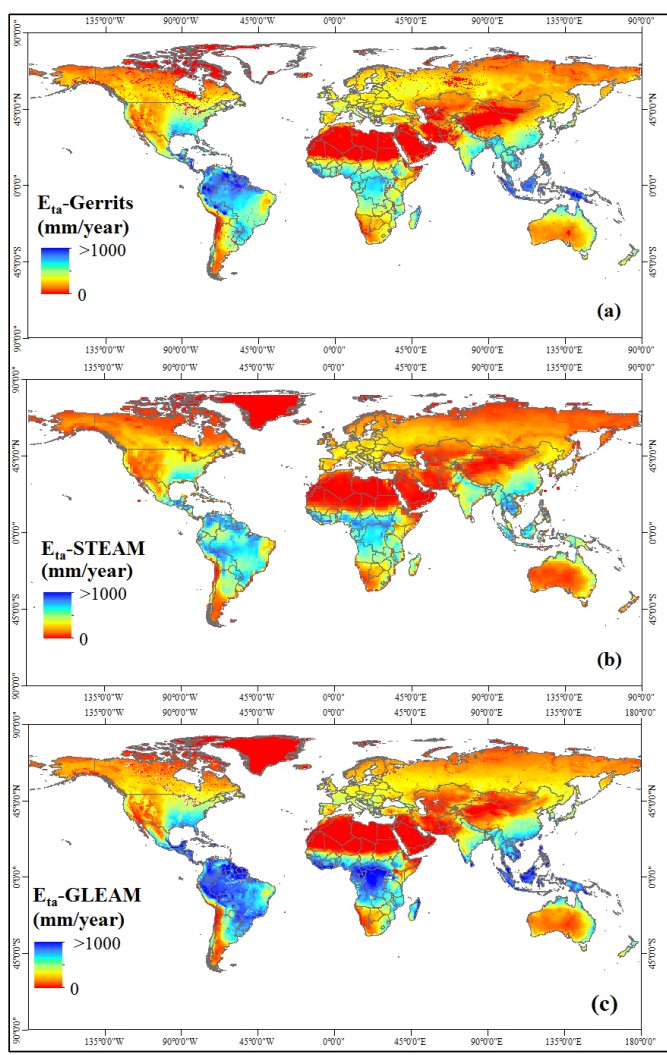

**Figure 6-** Simulated mean annual transpiration by (a) Gerrits' model, (b) STEAM and (c) GLEAM.





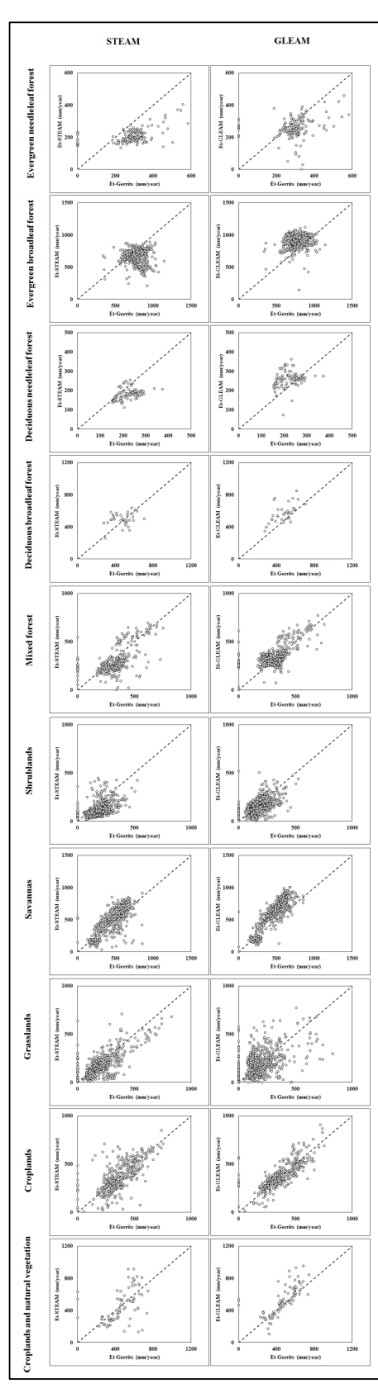

2 **Figure 7-** Scatter plot of annual transpiration estimated by Gerrits' model in comparison to
3 STEAM (left panel) and GLEAM (right panel) per land cover type.



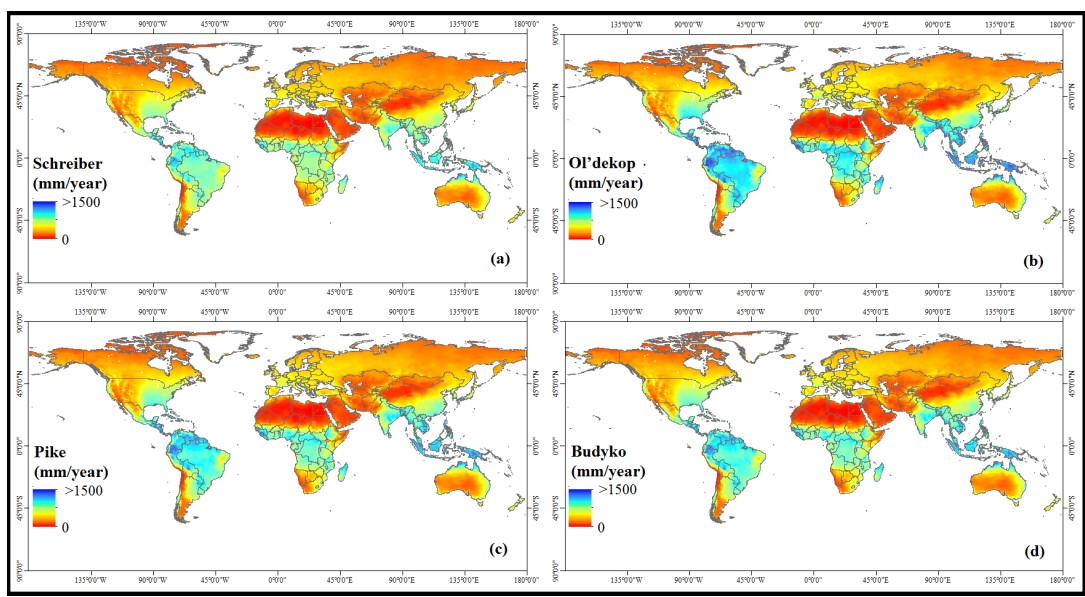

**Figure 8-** Global evaporation (mm year[-1]) estimated by Budyko curves: (a) Schreiber (1904), (b) Ol'dekop (1911), (c) Pike (1964), and (d) Budyko (1974).





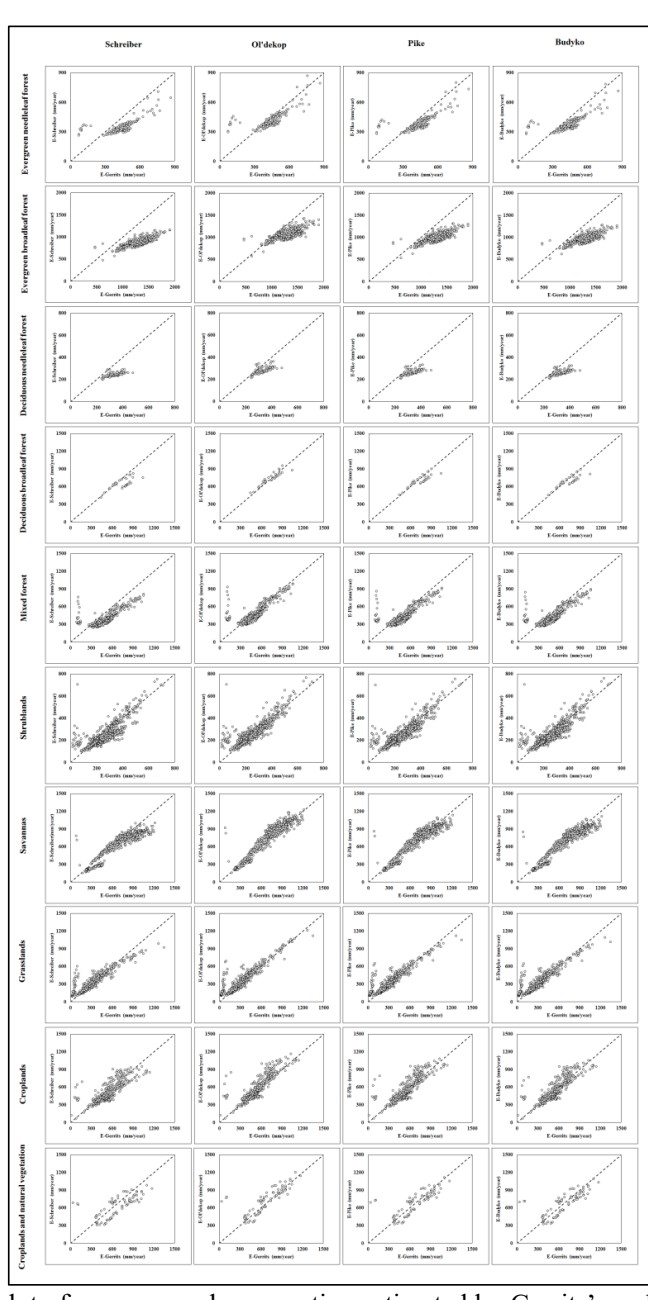

**Figure 9-** Scatter plot of mean annual evaporation estimated by Gerrits' model in comparison to Budyko curves: Schreiber (1904), Ol'dekop (1911), Pike (1964), and Budyko (1974) per land cover type.