# Peer review of "A simple global Budyko model to partition evaporation into 1 interception and transpiration 2 3 Ameneh Mianabadi1,2, Miriam Coenders – Gerrits2\*, Pooya Shirazi1, Bijan Ghahraman1, Amin Alizadeh1 4 5 1- Ferdowsi University of Mashhad, Mashh"

_Hydrology and Earth System Sciences, 2017_

## Referee Comment (RC1) · Anonymous Referee #1 · 22 Jun 2017

Annual evaporation (and interception and transpiration) at the global scale is estimated using the Gerrits' model, and compared with other data sources including Landflux-EVAL, STEAM and GLEAM. The estimated annual evaporation is also compared with one estimated by non-parametric Budyko equations. A global estimation of interception and transpiration is presented in this paper. The paper is well written, technically sound, and valuable (global interception and transpiration estimation). I provide the following comments for the authors to consider for revision.

It is not necessary to compare estimated evaporation with four non-parametric Budyko equations since all of them are curve fittings of observations and the uncertainty may be large for some areas. It is fine to compare it with one (e.g., Budyko equation). However, if it is possible, the authors may consider comparing the evaporation estimations

with parametric Budyko equations since the parameter of the Budyko equation can be linked to land surface properties such as LAI.

Since the Gerrits' model estimates both interception and transpiration, it is interesting to computing the ratio between interception and total evaporation, i.e., Ei/E. The parameter of the Budyko equation in Wang and Tang (2014, doi:10.1002/2014GL060509), $\varepsilon$, can be computed using precipitation, potential evaporation, and E. I am curious on the relation between Ei/E and $\varepsilon$. The authors may plot Ei/E versus $\varepsilon$. Is Ei equivalent to "initial evaporation" defined in the paper?

It seems that the Gerrits' model is a conceptual model (with inputs and parameters even though assumed to be constant) for computing interception and transpiration, but is not a simple Budyko equation.

A thorough proof-reading is recommended since there are some typos. For example, a few typos are listed below. Line 11 on page 2: "physical base" needs to be changed. Equation (1) itself does not explain any physical process. Line 26 on page 2: "a couple of" Line 4 on page 5: change "parameter" to "parameters"; "equation" to "Equation" Line 9 on page 5: any explanation on the value of b (=0.1)? Line 19 on page 5: "are limited" to "is limited" Line 19 on page 7: "aria" to "area"

---

## Referee Comment (RC2) · M. Roderick (Referee) · 23 Jun 2017

**Review of HESS manuscript #hess-2017-306**

Title:          A simple global Budyko model to partition evaporation into interception and transpiration

Author:        Mianabadi et al

This manuscript describes a global model of land surface evaporation that uses remotely sensed data to estimate various parameters of the original "Gerrits 2009 WRR" model.

The topic is important and within the scope of HESS.

One of the main reasons to use the "Gerrits 2009 WRR" model is that it was based on underlying reasoning that explicitly recognises the characteristic time scales for different processes, e.g. evaporation of water intercepted by the canopy has a characteristic time scale of order one day (see p. 3, lines 17-29 for relevant background).

However, as currently written I could not see how the manuscript followed this approach.

For example, as currently described (and I am not convinced the description is accurate), the daily potential evaporation is calculated as the annual potential evaporation divided by 365 (see p. 4, line 7). This means that every day of the year has the same potential evaporation. This means that there is no seasonal variation in the atmospheric demand imposed on the evaporation of water intercepted by the canopy.

The situation for transpiration is more or less the same. For example, the monthly transpiration threshold is calculated as the annual potential evaporation divided by 12 (see p. 5, lines 20-31). This means that every month of the year has the same transpiration threshold that is subsequently modified using the Novak and Jan (2005) formulation along with remotely sensed LAI.

While the focus of the paper is on annual totals, we know in terms of the underlying processes that seasonal variations are important. One justification of using the "Gerrits 2009 WRR" model is to explicitly recognise the time scales of the underlying processes. A seasonal time scale is also important and I do not see how potential evaporation being the same for every day/month of the year recognises the seasonal time scale? With that in mind, the two comments noted above seem major oversights.

Now I am not convinced that this is actually what has been calculated, but it is what has been described in the manuscript. If this is what has been calculated then the problem could be easily detected by examining some seasonal cycles in different regions. In fact an evaluation of the mean seasonal cycle of the new approach compared with other approaches could be considered useful in evaluating the new approach. However, all results (Figures 2-9, Tables 3-6) are presented as annual totals so the seasonal variations are not evident. Perhaps these seasonal cycles have been done?

Finally, in the formulation of the methodology you describe the basic equation, Eqn 2 (p. 3, line 6). After some discussion, there is an amended description for water bodies and for all other surface types (Eqn 3). However, the term Es from Eqn 2 has gone missing. The

implication is that the model does not consider evaporation from soil. I assume the model actually does calculate soil evaporation - it is just the overall description that does not consider this term.

I do not see that this manuscript was anywhere near ready for submission.

**Recommend: Reject.**

Other Comments:

1.      Table 2 has been a useful start. Rather than present this as a two column table of equations, why not have one column of equations, another for units and a final column for explanation of each term. Every variable in the paper needs to be here. As it currently stands some terms are explained in the table and some in the text and it is hard for a reader to follow all the different terms.

2.      Fig. 1c. The scale bar for LAI has a maximum at 60. This seems a little excessive? Is there a numerical problem in the figure?

Michael L. Roderick, 23-06-2017

---

## Author Comment (AC1) · 28 Jun 2017

Dear Michael Roderick,
We are very grateful for your detailed review on our manuscript, where we apply the Gerrits 2009 WRR-model on the global scale with parameters estimated from remotely sensed data sources.

The underlying reasoning of the Gerrits model is indeed to recognise the characteristic time scales of the different evaporation processes (i.e. interception daily and transpiration monthly). In Gerrits 2009 WRR (and in the current paper as well) this has been done by taking yearly averages for the interception ($D_{i,d}$, mm/day) and transpiration

threshold ($D_{t,m}$, mm/month) in combination with the temporal distribution functions for daily, and monthly (net) rainfall. Hence the seasonality is incorporated in the temporal rainfall patterns, and not in the evaporation thresholds. We agree that this is a limitation of the currently used approach and could be the focus of a new study by investigating how seasonal fluctuating thresholds (based on LAI and/or a simple cosines function) would affect the results. This could be a significant methodological improvement of the Gerrits-model, but will have mathematical implications on the analytical model derivation. For sure it will improve the monthly evaporation estimates, but we expect that the consequences at the annual time scale (which is the focus of the current paper) will be less severe.

Firstly, the consequences of using a constant interception threshold ($D_{i,d}$). We modelled the daily interception storage as the minimum of the storage capacity ($S_{max}$) and the daily potential evaporation ($E_{p,d}$), see Equation 15. For most locations $S_{max}$ is smaller than $E_{p,d}$ even if we consider a daily varying potential evaporation. $S_{max}$ (based on LAI) could also be changed seasonally, however many studies show that the storage capacity is not changing significantly between the leafed and leafless period (e.g., Leyton et al., 1967; Dolman, 1987; Rutter et al., 1975). Futhermore, Gerrits et al (2010) showed with a Rutter-like model that interception is more influenced by the rainfall pattern than by the storage capacity, which was also found by Miralles et al. 2010. Hence, in interception modelling, the value of the storage capacity is of minor concern.

We expect that the consequence of a constant transpiration threshold ($D_{t,m}$) is more important, especially in energy constrained areas. But in those, relatively wet, areas we will underestimate the transpiration in summer and overestimate it in winter, which will cancel out on the annual scale.

We agree with you this is a limitation of the Gerrits model and are grateful for your suggestion. In the revised manuscript we propose to add a detailed discussion on the implications of using a constant interception and transpiration threshold on the annual time scale.

Furthermore, it seems there is some confusion on whether soil evaporation ($E_s$) is taken into account in our model or not, since it is not considered in equation 3 anymore. We do consider 'soil evaporation', however we consider what many researchers call 'evaporation from the soil' as forest floor or ground interception (as part of total interception). Hence we define interception broader than only canopy interception. We define interception as "the amount of evaporation from any wet surface including canopy, floor, understory and the top layer of the soil, occurring within a day from the rainfall event". Hence interception evaporation is the fast feedback of moisture to the atmosphere originating from wet surface. Soil evaporation is then defined as "evaporation of soil moisture that is connected to the root zone (De Groen and Savenije, 2006)" and is therefore different from evaporation of the top layer of the soil (several millimetres of soil depth, which is here considered as part of the interception evaporation). Gerrits et al (2009) assumed that evaporation from soil moisture is negligible (or can be combined with interception evaporation). As a result equation 2 becomes E=Ei+Et for land surfaces, where Ei is direct feedback of moisture stored on vegetation, ground, and top layer, while Et is evaporation from soil moisture storage in the root zone, which includes soil moisture that evaporates directly to the air by capillary rise.

We understand your confusion, and therefore propose to replace the first paragraph of the Methodology section with the definitions as given above. We hope these definitions clarify your question.

Lastly, we like your suggestion to add a list of symbols. We agree that this will help the reader understanding the paper more easily. Also the issue with Figure 1c (LAI) will be fixed. The maximum LAI is indeed not 60, but 6.

References:
Dolman, A. J. (1987), Summer and winter rainfall interception in an oak forest: Predictions with an analytical and a numerical simulation model, J. Hydrol., 90, 1–9

Gerrits, A. M. J., Savenije, H. H. G., Veling, E. J. M. and Pfister, L.: Analytical derivation of the Budyko curve based on rainfall characteristics and a simple evaporation model, Water Resources Research, 45, 2009.

Gerrits, A. M. J., Pfister, L. and Savenije, H. H. G.: Spatial and temporal variability of canopy and forest floor interception in a beech forest, Hydrological Processes, 24(21), 3011–3025, doi:10.1002/hyp.7712, 2010.

Leyton, L., Reynolds, R. C. and Thompson, F. B.: Forest hydrology, edited by W. E. Sopper and H. W. Lull, pp. 163–179, Pergamon Press, Oxford., 1967.

Miralles, D. G., Gash, J. H., Holmes, T. R. H., de Jeu, R. A. M. and Dolman, A. J.: Global canopy interception from satellite observations, J. Geophys. Res., 115(D16), D16122–, 2010.

Rutter, A. J., Morton, A. J. and Robins, P. C.: A predictive model of rainfall interception in forests. II Generalization of the model and comparison with observations in some coniferous and hardwood stands, Journal of Applied Ecology, 12, 367–380, 1975.

De Groen, M. M. and Savenije, H. H. G.: A monthly interception equation based on the statistical characteristics of daily rainfall, Water Resources Research, 42, doi:10.1029/2006WR005013, 2006.

---

## Author Comment (AC2) · 11 Aug 2017

The authors would like to express their sincere gratitude to the Anonymous Referee #1 for his/her useful comments. The reactions to the comments are as follows.

**Comment 1:**

It is not necessary to compare estimated evaporation with four non-parametric Budyko equations since all of them are curve fittings of observations and the uncertainty may be large for some areas. It is fine to compare it with one (e.g., Budyko equation). However, if it is possible, the authors may consider comparing the evaporation estimations with parametric Budyko equations since the parameter of the Budyko equation can be linked to land surface properties such as LAI.

**Reaction:**

We will skip the Schreiber, Ol'dekop and Pike equations and only keep the original Budyko curve. Furthermore, we will try the parametric equation of Fu based on the empirical equation of $\omega = 2.36M + 1.16$ (with $\omega$: Fu parameters, $M = \frac{NDVI - NDVI_{min}}{NDVI_{max} - NDVI_{min}}$) (Li et al., 2013).

**Comment 2:**

Since the Gerrits' model estimates both interception and transpiration, it is interesting to computing the ratio between interception and total evaporation, i.e., Ei/E. The parameter of the Budyko equation in Wang and Tang (2014, doi:10.1002/2014GL060509), $\varepsilon$, can be computed using precipitation, potential evaporation, and E. I am curious on the relation between Ei/E and $\varepsilon$. The authors may plot Ei/E versus $\varepsilon$. Is Ei equivalent to "initial evaporation" defined in the paper?

**Reaction:**

This comment is an interesting suggestion. We will provide the global maps of Ei/E and Et/E in the manuscript and then we will make a rough link to $\varepsilon$ to see if there is any relation between Ei/E and $\varepsilon$.

**Comment 3:**

It seems that the Gerrits' model is a conceptual model (with inputs and parameters even though assumed to be constant) for computing interception and transpiration, but is not a simple Budyko equation.

**Reaction:**

The Gerrits' model is indeed a conceptual model for computing interception and transpiration. The 'output' of this model is an equation, which provides similar output as other Budyko equations. We do agree that the Gerrits' model at the annual time scale is not a simple equation, since it uses complex mathematical functions. Therefore we decided to remove the word 'simple' from the title and call our model a "Budyko model"

.

**Comment 4:**

A thorough proof-reading is recommended since there are some typos. For example, a few typos are listed below.

**Reaction**

Thank you for finding these typos. We will correct all the typos and do a careful proofreading. Furthermore, "$b$=0.2" gave the best global results for all land classes. We will mention this in the revised manuscript.

**References**

Li D, Pan M, Cong Z, Zhang L and Wood E (2013) Vegetation control on water and energy balance within the Budyko framework. Water Resources Research *49*(2): 969–976